# Closing the Loop on LIB Waste: A Comparison of the Current Challenges and Opportunities for the U.S. and Australia towards a Sustainable Energy Future

Gavin E. Collis [1,]*, Qiang Dai [2], Joanne S. C. Loh [3], Albert Lipson [2], Linda Gaines [2], Yanyan Zhao [4] and Jeffrey Spangenberger [2,]*

1  CSIRO Manufacturing, Research Way, Clayton, VIC 3169, Australia
2  Argonne National Laboratory, Lemont, IL 60439, USA
3  CSIRO Mineral Resources, 7 Conlon Street, Waterford, WA 6152, Australia
4  CSIRO Energy, Research Way, Clayton, VIC 3169, Australia
*  Correspondence: gavin.collis@csiro.au (G.E.C.); jspangenberger@anl.gov (J.S.)

**Abstract:** Many countries have started their transition to a net-zero economy. Lithium-ion batteries (LIBs) play an ever-increasing role towards this transition as a rechargeable energy storage medium. Initially, LIBs were developed for consumer electronics and portable devices but have seen dramatic growth in their use in electric vehicles (EVs) and via the gradual uptake in battery energy storage systems (BESSs) over the last decade. As such, critical metals (Li, Co, Ni, and Mn) and chemicals (polymers, electrolytes, Cu, Al, PVDF, $LiPF_6$, $LiBF_4$, and graphite) needed for LIBs are currently in great demand and are susceptible to global supply shortages. Dramatic increases in raw material prices, coupled with predicted exponential growth in global demand (e.g., United States graphite demand from 2022 7000 t to ~145,000 t), means that LIBs will not be sustainable if only sourced from raw materials. LIBs degrade over time. When their performance can no longer meet the requirement of their intended application (e.g., EVs in the 8–12 year range), opportunities exist to extract and recover battery materials for re-use in new batteries or to supply other industrial chemical sectors. This paper compares the challenges, barriers, opportunities, and successes of the United States of America and Australia as they transition to renewable energy storage and develop a battery supply chain to support a circular economy around LIBs.

**Keywords:** battery value chain; lithium-ion battery recycling; black mass; critical materials; sustainability; circular economy; net-zero emissions; government policies; global standards





## 1. Introduction

Lithium-ion batteries (LIBs) are ubiquitous in technologies required for decarbonization, such as electric vehicles (EVs) and battery energy storage systems (BESSs), and are, thus, crucial to a net-zero-emissions future [1]. Some common EVs are battery electric vehicles (BEVs, battery and electric motor) and plug-in hybrid electric vehicles (PHEVs, has both a battery-driven electric motor and an internal combustion engine (ICE)). Although battery recycling has received much attention from the research community in recent years, commercialization of recycling technologies remains challenging, and recycling infrastructure is still lacking in many countries; there is still significant room for improvement [2–11]. This paper discusses how geography, population, government, and industrial perspectives of Australia and the United States (U.S.) influence the barriers, technical challenges, opportunities, and risks for LIB recycling. This understanding can help diverse stakeholders plan and implement solutions to address LIB recycling and battery supply chain gaps in both nations.

## 2. Results and Discussion

*2.1. Background—Population Size and Distribution and Land Area*

The U.S has 50 states. Based on the 2020 census, the U.S. has a population of 331.45 million and a land area of 3.53 million square miles [12]. Figure 1 shows the population distribution. Figure S1 shows a map of the population density for each state in the U.S. [13].

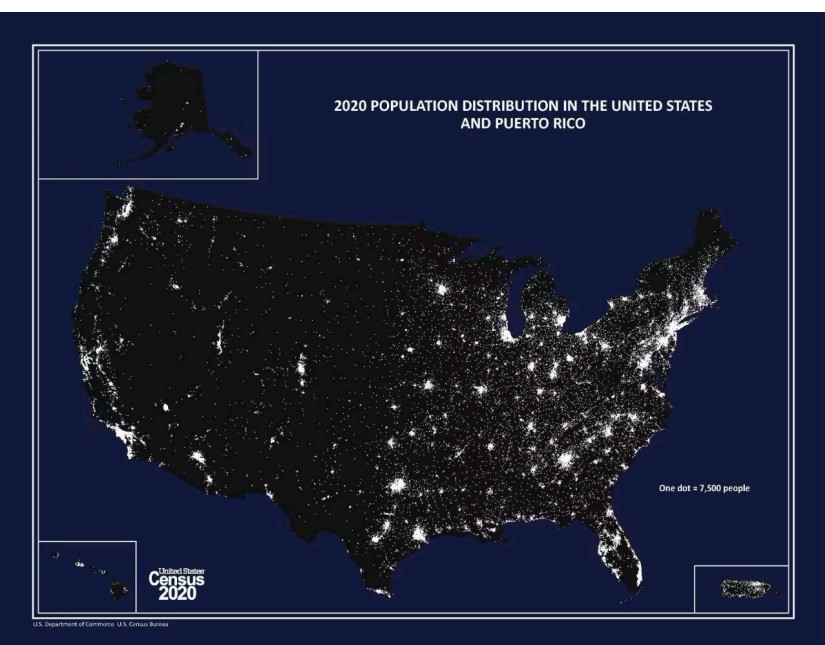

**Figure 1.** 2020 U.S. population distribution (Reproduced from U.S. Census Bureau [13]).

In contrast, Australia is made up of six states and two territories (Figure 2) with a land area of 7.688 million km$^2$ (~3.0 million square miles) [14]. From the 2021 Census by the Australian Bureau of Statistics, Australia's population was reported as 25,978,935 [15], mainly concentrated around the eight cities located along the coast (Figure 2) [14].

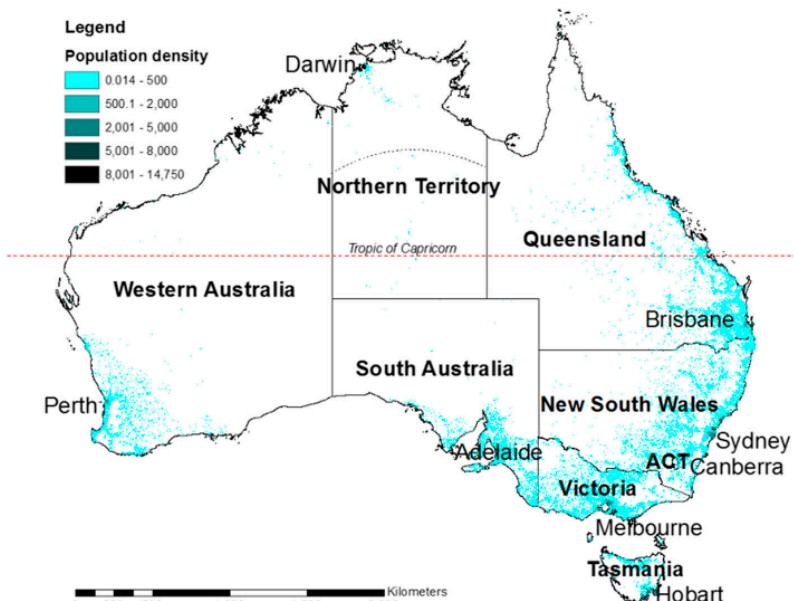

**Figure 2.** Map of Australia showing population density (Reproduced from Australian Bureau of Statistics [16]).



It is plain to see that while the two countries are broadly similar in land dimensions (i.e., distance (by air) between Perth and Brisbane is 2241.03 mi (3606.59 km) and distance between Los Angeles and New York is 2445.55 mi (3935.74 km), Australia's population density is less than 10% of that of the U.S. This is particularly evident in the comparison of Texas, the largest state in the continental U.S., with Australia's biggest state, Western Australia; Texas is only 0.28 times (by area) the size of Western Australia. However, the population of Texas is 30.01 million (2022), which is approximately 10 times the population of Western Australia (2.83 million people in 2022).

### 2.2. Battery Use—EVs and BESSs

LIBs play an increasingly important role in the transition towards clean and low-carbon technologies. As of 31st December 2021, 2.24 million passenger BEVs (powered by LIBs) were registered in the U.S. [16]. California alone accounts for 0.878 million registrations [16], and the distribution of BEV registrations largely mirrors that of the population distribution. According to the International Energy Agency, the U.S. BEV stock may reach 20–40 million by 2030, depending on the projection scenario [17].

Compared with BEVs, LIB deployment for BESSs in the U.S. is small in volume. As of 2021, operational BESSs in the U.S. based on LIBs have a combined energy capacity of 10.2 GWh, which is equivalent to ~0.13 million Tesla Model 3 units. Most of the existing BESS capacity is concentrated in California, Massachusetts, New York, and Texas [18]. Based on U.S. Energy Information Administration projections, the U.S. battery storage capacity is expected to grow to 235 GWh by 2050 [19].

Unlike the U.S. and other countries (see Section 2.4 later), Australia's uptake of EVs has been slow and would seem to be constrained by the vast distances and low population densities across the country that connect major cities and regional centres (see SI, Section S2). Financial incentives to promote EV uptake were only recently passed by the Australian Federal Government in November 2022 (while state governments have enacted various levels of incentives over the past five years). Interestingly, Australia considered PHEVs as early as the mid-1990s. The Commonwealth Scientific and Industrial Research Organisation (CSIRO) commenced a collaboration with General Motors Holden (GMH), a subsidiary company of General Motors in the U.S. [20], to develop a hybrid ICE with an electric motor and lead–acid battery storage. The ECOmmodore (Figure S2) [21] was aimed towards reducing hydrocarbon and greenhouse gas (GHG) emissions. Although publicly launched in May 2000, the vehicle never entered commercial production, nor did it stimulate an EV market in Australia.

BEV Tesla Model S sales began in Australia in 2014, shortly followed by the BMW i3. 2023 data by the EV Council of Australia [22,23] showed that BEVs dominate the market share of EVs (Figure S3). Despite commercial EVs entering the Australian market from 2009 to 2018, there was very little uptake compared to other countries, suggesting other factors were contributing to the low sales. To complicate matters, from the mid-1980s through to the mid-2010s, Australia's auto industry experienced significant changes that ultimately led to the demise of car manufacturing (Holden, Toyota, Mitsubishi, and Ford) in Australia by 2017, driven by domestic and global market forces and changes to Federal Government policies [24]. In contrast, several U.S. car manufacturers started to transition to EVs, PHEVs, and BEVs, whilst selling more efficient ICEs, which has helped support and grow a domestic LIB supply chain. BEV usage is centralized around city areas where charging infrastructure is being continually introduced and upgraded. PHEVs that accommodate both the short, daily commute in high populated areas and longer drives to regional areas with fewer charging stations may ease the transition to electrified transport by addressing "range anxiety" in the short term.

In contrast, BESSs have seen continual growth and expansion across Australia [25]. The Australian BESS market currently consists of several players, such as Pacific Green Technologies Group, LG Energy Solution Ltd., Tesla Inc., Century Yuasa Batteries Pty Ltd., and EVO Power Pty Ltd., but is changing rapidly with the uptake of different energy

storage technologies [26,27]. However, the most popular and publicized effort was the 100-day challenge set by Elon Musk to build a 100 MW system, the world's largest LIB BESS at the time, at Hornsdale in South Australia (Figure S4) [28]. The Hornsdale BESS addressed the problems associated with severe electricity blackouts in 2016 and 2017 and demonstrated that renewable energy could be used as a back-up energy source.

Not surprisingly, this success has stimulated other LIB BESS projects in other states, such as the Victorian Big Battery program, a 300 MW/450 MWh BESS located in Moorabool, in 2021 [29]. The system stores abundant renewable energy during the day and also excess electricity from neighbouring state New South Wales during peak times, especially during hot summer weather. The project was developed by the local subsidiary of France-headquartered energy provider Neoen and uses Tesla BESS equipment (Figure S5).

Based on information from the Clean Energy Council, there are 34 BESS projects that are currently or soon to be under construction (Figure 3), ranging in scale from small utilities to very large systems. Some of these are summarized in the Table S1 in the Supplementary Information [30].

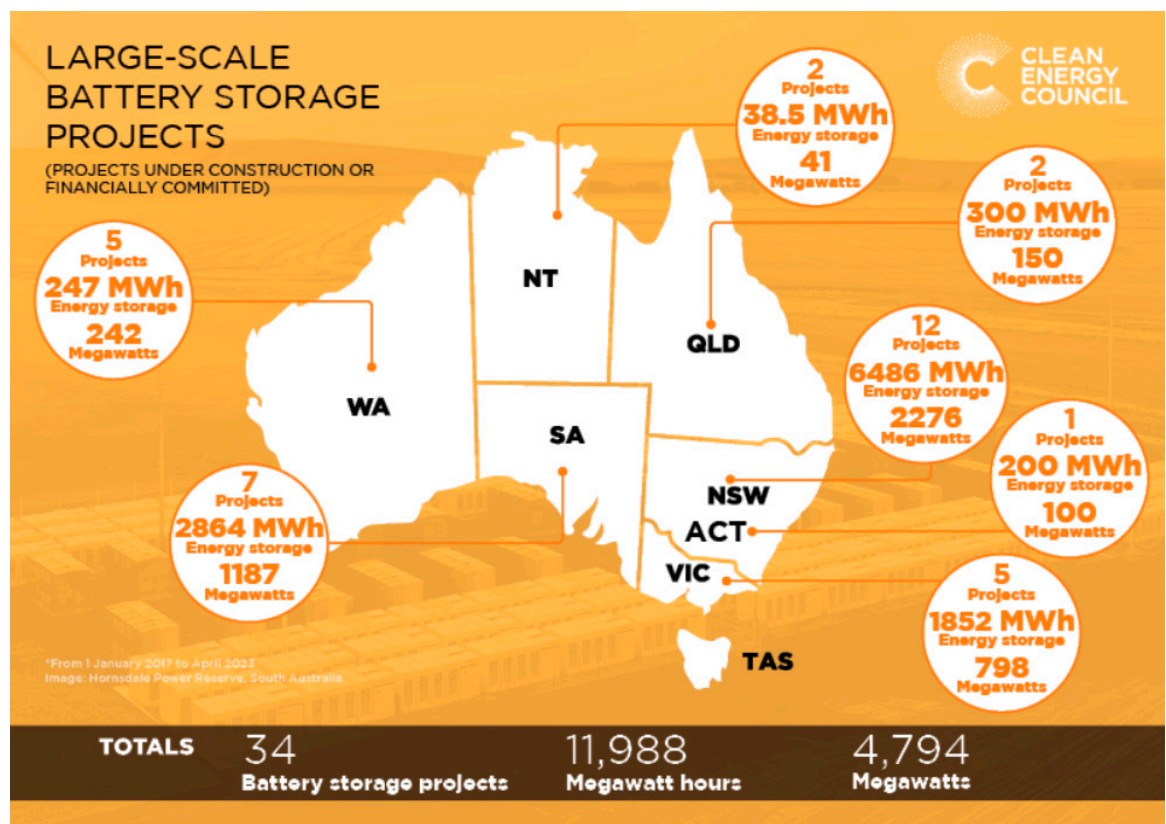

**Figure 3.** Large-scale battery storage projects in Australia as of 2022 (Reproduced from [31]).

In addition, the Australian government has agreed to an AUD 117.5 million investment, managed and funded through the government-owned Australian Renewable Energy Agency's (ARENA) Large-Scale Battery Storage Funding Round, as part of AUD 1.8 billion worth of projects (late 2022). These projects will deliver eight large-scale batteries with a combined 2 GW/4.2 GWh of storage capacity, resulting in a tenfold increase in grid-forming electricity storage capacity (Figure 4) [32,33]. This grid-scale infrastructure will maintain grid stability without the need for coal and gas generators, enabling Australia's goal towards an electricity grid that can support 100% renewable energy.

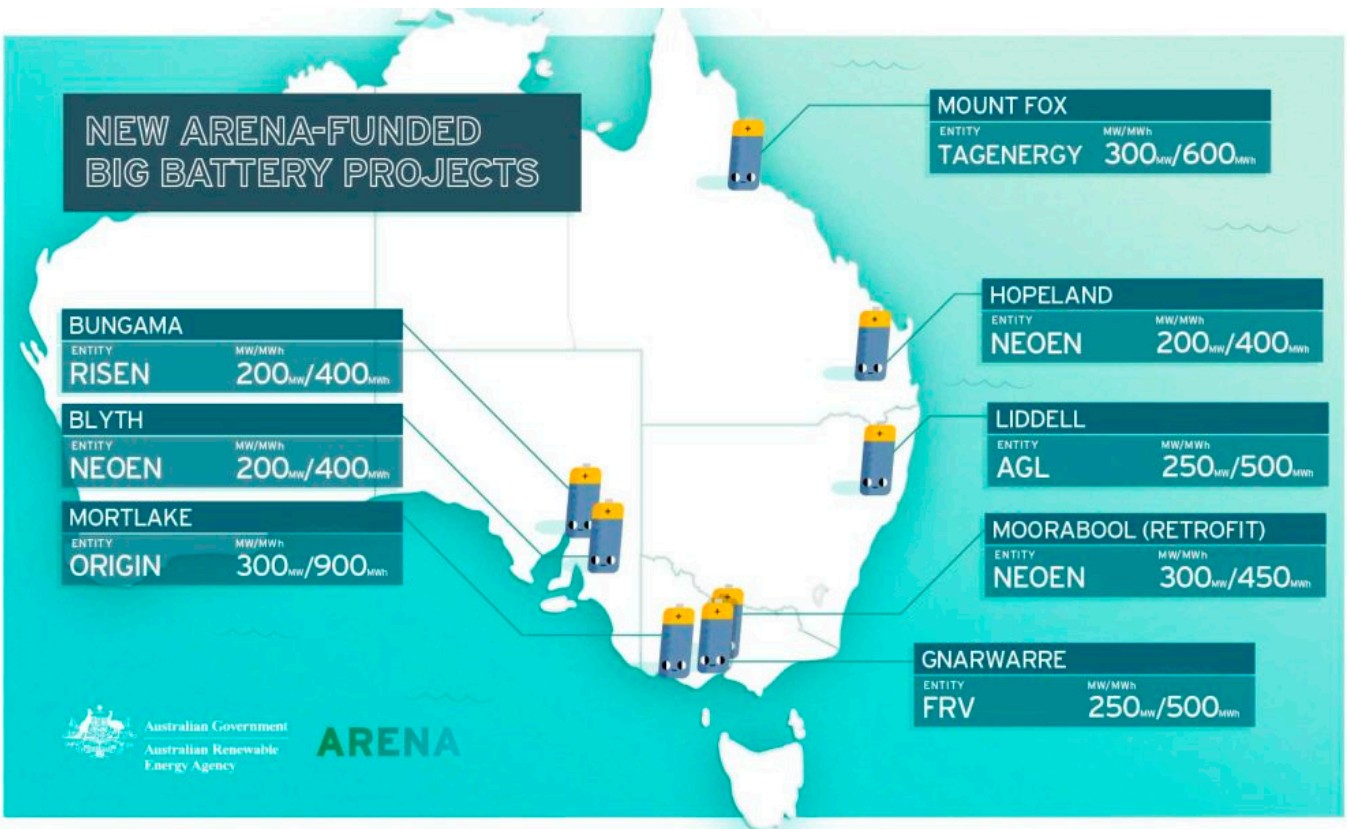

**Figure 4.** Planned large BESSs, funded by ARENA, to be constructed around Australia (Reproduced from [32,33]).

*2.3. LIB Supply Chain—From Raw Materials to End of Life (EOL)*

2.3.1. Sourcing Raw Minerals and Battery Chemicals

Supporting the rapid growth of LIB use in the U.S. will require a significant quantity of battery minerals, such as lithium (Li), nickel (Ni), and cobalt (Co), for NMC cathode battery chemistry. Currently, the U.S. does not have sufficient domestic supply of these materials to meet projected demands. Table 1 summarizes U.S. and global reserves and resources for Li, Ni, and Co.

**Table 1.** U.S. and Global Reserves and Resources of Battery Minerals (in metric tons of contained metal).

| Metal | USA | | World | |
|---|---|---|---|---|
| | Reserve | Resource | Reserve | Resource |
| Li [34] | 750,000 | 9,100,000 | 22,000,000 | 89,000,000 |
| Co [35] | 69,000 | 1,000,000 | 7,6000,000 | 145,000,000 [a] |
| Ni [36] | 340,000 | N/A | >95,000,000 | 300,000,000 [b] |

[a]: 25 million metric tons (t) in terrestrial resource and 120 million t in sea nodules; [b]: Terrestrial resource only.

The lack of domestic reserves of battery minerals and recent supply disruptions during the COVID pandemic have prompted the battery industry to move towards vertical integration and explore unconventional sources to secure material supplies. The Inflation Reduction Act (IRA), which was passed in August 2022, provides bonus credits to batteries manufactured or assembled in North America and/or containing critical minerals extracted or processed in the U.S. or countries with a free trade agreement with the U.S. [37]. The act has encouraged further expansion of production capacity of battery minerals in the U.S. As a result, 13 new production facilities of battery-grade materials have been planned in the

U.S. as of August 2022, according to the NAATBatt LIB Supply Chain Database (hereinafter referred to as NAATBatt database) [38]. Table 2 summarizes current and planned battery materials production capacities in the U.S.

**Table 2.** U.S. Current and Planned Battery Materials Production Capacities (Adapted from NAATBatt database) [38,39].

| Mining and Refining | | | | |
|---|---|---|---|---|
| | **Number of Facilities** | | **Annual Production Capacity (t)** | |
| | Current | Planned | Current | Planned |
| Li | 2 | 4 | 940 * | 40,925 * |
| Ni | 2 | - | 19,000 * | - |
| Co | - | 2 | - | 76,915 * |
| Graphite | - | 2 | - | 7400 |
| **Battery-Grade Materials Production** | | | | |
| | **Number of Facilities** | | **Annual Production Capacity (t)** | |
| | Current | Planned | Current | Planned |
| Li | 3 | 11 | N/A | 39,958 * |
| Ni | - | 1 | - | 62,700 * |
| Co | - | 1 | - | 5500 * |
| Graphite | 9 | 9 | 7000 | 145,068 |
| Cathode Active Material | 3 | 4 | 7000 | 40,000 |
| Silicon | 3 | 6 | N/A | 7400 |

*: t of contained metal.

Australia has resources of all eight minerals needed for common commercial LIBs (lithium, cobalt, nickel, graphite, manganese, aluminium, iron (steel), and copper). The Australian government, through Geoscience Australia [40], considers 26 resource commodities to be critical minerals, taking into account both Australia's and selected partner countries' needs (see Table S2 and Figure S6). Table 3 summarizes Australia's potential for supplying EV battery materials and Figure S7 shows a map of the location of major mining and mineral deposits of other valuable base metals.

**Table 3.** Australia's Natural LIB Resources and Production Compared Against Global Production (Reproduced from [41]).

| Material | Australian Economic Demonstrated Resources (2020) | Australia Production (2020) | World Mine Production (2020) |
|---|---|---|---|
| Lithium | 6174 kt | 40 kt | 82 kt |
| Cobalt | 1495 kt | 5.6 kt | 135 kt |
| Nickel | 21,400 kt | 170 kt | 2500 kt |
| Manganese ore | 276,000 kt | 4800 kt | 17,200 kt |
| Graphite | 7970 kt | 0 | 1100 kt |

Since 2015, a variety of government-funded reports [42–47] have investigated the importance and role that Australia can play in supplying critical minerals to domestic and international markets for a variety of renewable energy technologies. The key conclusions

from these reports, as well as from more recent media releases relating to supply chains of batteries or LIBs, include the following:

-   Australia has abundant natural raw materials for manufacturing batteries in Australia; however, almost all of these are exported overseas as unprocessed ores;
-   Australia is developing upstream industries and supply chains with a focus on adding value by processing ores to battery-grade materials domestically rather than continuing to ship raw or concentrated ores overseas [47];
-   The global demand for battery materials has also seen a shift in the importance of LIB waste. In 2017, the domestic production of black mass was viewed as the solution to prevent EOL LIB waste ending up in landfill and had a value averaging around AUD 1000–2000/tonne. Today, the demand and price for "high-quality" black mass prices are starting to surge to between ~AUD 5000 and 10,000/tonne in some parts of the world as countries see the commercial opportunity to enter and develop LIB supply chains [47–51];
-   If EOL LIB battery chemistry can be easily sorted, there are opportunities for Australia to leverage off its specialized expertise in mining and processing of ore bodies and also consider the use of high-quality black mass as feedstock in their process [47]. This would add significant value to their industry portfolio by using a "refined" feedstock and would increase revenue by adding value to their current ore pipeline, as suggested by initial technoeconomic studies;
-   Capabilities for processing critical minerals in Australia are on the increase; a joint venture between Tianqi Lithium Corp. and IGO produced Australia's first battery-grade lithium hydroxide in commercial quantities at the Kwinana Lithium Hydroxide Refinery in May 2022 [52]. Another lithium hydroxide refinery, owned by a joint venture between Albemarle (Kemerton, WA, Australia) and Minerals Resources Ltd (Osborne Park, WA, Australia) is currently under construction [53]. BHP Nickel West produced Australia's first nickel sulphate in October 2021 [54];
-   Through the Future Battery Industries Collaborative Research Centre (FBICRC), Australia's first small-scale pilot plant for cathode precursor chemicals commenced production in July 2022 and aims to develop the technology and capability to establish a cathode precursor manufacturing industry [55];
-   Recent support [56] by previous and incumbent Australian governments has resulted in large funding grants for the development of critical mineral projects along the value chain (see below); however, there are no specific policies that provide or create the "market pull" or need for battery specific products, services, or jobs.

The impacts from the COVID pandemic have highlighted the fragility of global supply chains, prompting significant discussions and efforts to develop sovereign capability related to energy security for Australia and U.S. Efforts to accelerate development and investment into critical minerals projects along supply chains in Australia have largely been through Federal Government loans and grants [57]. Through Export Finance Australia (in conjunction with other funds), flexible finance loans have been offered to companies including Pilbara Minerals Ltd (West Perth, WA, Australia) (AUD 125M, expansion of Li mining and beneficiation operations). The AUD 2B Critical Minerals Facility (set up by the previous government) has issued (conditional) loans to Ecograf Ltd. (West Perth, WA, Australia) (AUD 40M, expansion of its Australian Battery Anode Material Facility (graphite)) and Renascor Resources Ltd. (Adelaide, SA, Australia) (AUD 185M, development of the Siviour Graphite Project). The Modern Manufacturing Initiative (MMI) (AUD 243M, 2020-2022) [58] provided grants to support the scaling-up, collaboration, and commercialization of critical minerals projects. The Critical Minerals Development Program, the most recent funding program, provides up to AUD 50M to help progress early-to-mid-stage projects that will strengthen Australia's sovereign capability in critical minerals towards financing and production [59]. Similarly, the U.S. grows the supply chain mainly through incentives (through the IRA as discussed earlier) and funding from the Bipartisan Infrastructure Law (BIL). Notably, in 2022, the Department of Energy (DoE) selected 20 companies to receive AUD 2.8B

of BIL funding for battery materials processing and battery manufacturing [60]. Moreover, after the IRA expanded eligibility for financing through the DoE's Loan Programs Office, the office has offered conditional commitments totalling AUD 13.1B to five companies, including AUD 9.2B to BlueOval SK for EV battery manufacturing, AUD 850M to KORE Power for EV and BESS battery manufacturing, AUD 2B to Redwood Materials and AUD 375M to Li-Cycle for battery material recovery and remanufacturing from recycling LIBs, and AUD 700M to Ioneer Rhyolite Ridge for lithium carbonate production [61].

Although there is growth in LIB use in Australia, there are limited downstream and value-adding options for raw resources, which includes the lack of an ICE or EV car manufacturing industry and battery cell manufacturers. The absence of a resilient manufacturing industry in Australia has slowed the development of a domestic battery supply chain, and the priority continues to be with the supply of raw materials to meet the growing international demand. However, international industries and countries requiring battery minerals and chemicals are now targeting Australia's abundance of minerals for use in the renewable energy sector [47], which includes EVs and BESSs, and there has been an increase in investment for infrastructure to support the establishment of domestic supply chains for the battery market in Australia.

### 2.3.2. Battery Manufacturing and Assembly

A study conducted by Argonne National Laboratory showed that of the total battery capacity in EVs sold since 2010, over half of the cells were manufactured within the U.S., while ~90% of the packs were assembled domestically [62]. Based on the NAATBatt database, the U.S. had an annual battery manufacturing capacity of ~80 GWh in 2022. Taking into account all of the announced new plants and expansions, including the recent announcement of the expansion of Tesla's gigafactory in Nevada, the capacity is expected to grow to 830 GWh in 2030 [63]. In comparison, current U.S. battery assembly capacity totals ~55 GWh per year, with ~560 GWh per year of capacity planned or under construction [38].

In contrast, Australia's relatively small population, vast unpopulated area, lack of a domestic auto manufacturing industry, and the associated lower demand for batteries, combined with the lack of domestic supply of battery chemicals and components, have limited the growth of a large-scale local battery manufacturing industry. However, there are companies in the LIB assembly sector (using battery components sourced from overseas) producing bespoke batteries for niche applications (e.g., Lithium Batteries Australia) [64]. Energy Renaissance [65] and Recharge [66] are committed to manufacturing and assembling local batteries with Australian-sourced materials [57].

### 2.3.3. Current Status of LIB Recycling Globally

To date, EOL LIBs can be categorized as originating from three types of waste streams: (1) consumer electronics including phones, laptops, and small hand-sized devices, (2) EVs, and (3) BESSs [67,68]. In addition, scrap generated from battery and pack manufacturing activities could also be considered as waste. Some of the current challenges associated with recycling LIBs are: (1) the variety of battery chemistries that make separation of individual metals during the recycling process more difficult, (2) the significant variability in size, shape, and different architectures of commercial LIBs which makes disassembly of individual cells difficult (i.e., battery cells, modules, and packs not designed for recycling), (3) the lack of adequate labelling to identify the type of battery chemistry, (4) safety aspects associated with the transportation and storage of EOL batteries, (5) discharge of EOL batteries to minimize fires, (6) wastes and emissions generated from the recycling process, and (7) changing or lack of suitable global and national guidelines and incentives around who is responsible and how to best address this growing waste stream [69].

Over the last few years, there have been numerous journal publications, review articles, and reports highlighting the current LIB recycling processes and potential options for improvements to reduce energy consumption and gas emissions which are well documented [70–77]. The two most common industrial methods for recycling LIBs are

pyrometallurgy and hydrometallurgy. More recently, direct recycling [78,79] and biometallurgy [80] are being developed on greater scales and their effectiveness have yet to be addressed. The aim of this section is to highlight some of the current issues and efforts towards improving these processes so that energy, waste, and emissions can be reduced to establish circular and sustainable economies.

The pyrometallurgy process is recognized as the simplest approach to recycling LIB waste [81–85]. In essence, it is a process that uses a very high temperature to extract metals or other compounds, producing a mixed metal alloy and a slag which can undergo further chemical processing. In conventional pyrometallurgical processing, waste LIBs are initially roasted at approximately 1000 °C, which results in the decomposition of the plastic, electrolyte, electrolyte salt, and separator components, producing toxic emissions such as dioxins, furans, fluorophosphorous materials (from electrolyte salt), hydrofluoric acid, hydrochloric acid, $SO_x$, and other acidic gases. After this stage, the battery mass is halved. The remaining materials are then heated to 1400–1700 °C to form a metal alloy and slag. The overall recovery rate of battery materials by pyrometallurgy is low [84,85].

The pyrometallurgical process requires additional capture and treatment infrastructure to manage the emissions generated during the process. In addition, the process has strict operational requirements for equipment to withstand the high temperatures and corrosive gas emissions, which leads to high, ongoing maintenance costs [85]. While technically feasible, the cost and energy requirements associated with recovering lithium from the slag mean that it is generally not economically viable [86]. The waste slag also contains large amounts of alumina, lithium oxide, and manganese dioxide, with some directly used as construction material, and the rest often still requires waste disposal [87].

The most attractive benefit of pyrometallurgical processing is its robustness and direct applicability to all LIBs, regardless of shape, size, or battery chemical composition. This eliminates the labour-intensive disassembly processes at the pack level and the shredding infrastructure typically needed for other recycling processes [85]. A number of global recycling companies have adopted pyrometallurgical processing in several countries, such as Umicore (Belgium), Inmetco (U.S.), Valdi (France), Redux (Germany), Dowa Eco-System (Japan), and Sumitomo/Sony (Japan) [67,76]. As various mining operations in Australia take ore bodies and refine the main metal species, the Canadian battery recycler Li-Cycle and their mining partner Glencore are investigating the feasibility of sourcing LIB black mass from Europe and processing the material using their pre-existing metallurgical complex to complement their supply of Ni, Co, and Li from raw minerals [88].

Hydrometallurgy is the dominant battery recycling technique in Asia, while pyrometallurgy is more commonly used in the European Union, either alone or in combination with hydrometallurgy, for battery recycling [89]. Currently, there are only a few countries that chemically process LIB black mass (i.e., pre-processed material obtained from shredding of LIB) to extract valuable metals, such as nickel, cobalt, manganese, lithium, copper, and aluminium, with varying degrees of efficiency using the hydrometallurgy process.

Commercialized hydrometallurgy recycling processes typically consist of leaching, solvent extraction, and precipitation. They are used to extract valuable minerals from a mixture of cathode and anode materials which are often obtained from a series of physical separation and/or thermal treatment steps on batteries and/or manufacturing scrap. Typical outputs from hydrometallurgical recycling processes include nickel, manganese, cobalt salts (where Ni, Mn, and Co are recovered separately), or nickel manganese cobalt precursors (where Ni, Mn, and Co are recovered together, most likely in the form of an NMC hydroxide). In addition, hydrometallurgical processes can recover scrap copper and aluminium and may also produce a lithium compound.

Compared with pyrometallurgical recycling, which burns off the organic compounds present during the smelting process, generating gaseous emissions, hydrometallurgy recycling does not need to deal with gas treatment unless upstream pre-processing involves thermal treatment. On the other hand, hydro-based recycling uses strong acids (for leaching), strong alkalis (for impurity removal and/or precursor production), and organic

solvents (for solvent extraction) and, therefore, requires wastewater treatment. The feedstock for leaching is black mass; efficient pre-processing of batteries and/or manufacturing scrap is a prerequisite for a successful hydrometallurgy process to sort the battery chemistry. Hydrometallurgical recycling may be an economically viable solution for cathode chemistries containing Co and Ni, especially when their prices are high and/or the feedstock is manufacturing scrap and features relatively high contents of Co and Ni. For batteries with lithium iron phosphate (LFP) or lithium manganese oxide (LMO) cathodes, however, the only valuable material hydrometallurgy recycling can recover is Li, which makes the economics a great deal more challenging.

Direct recycling methods, wherein the cathode material is preserved through the recycling process, may be able to overcome some of the economic challenges of recycling LFP and LMO. This is due to the significant processing cost needed to transform the raw materials into the final cathode material. A direct recycling process retains this value and, thereby, can significantly increase the revenue of the recycling process. However, direct recycling faces challenges in creating materials with consistent quality that will be accepted by the battery industry. Furthermore, there will be significant challenges in achieving commercial scale as large quantities of cathode materials are needed for qualification activities prior to being sold. For hydrometallurgical and pyrometallurgical approaches, only purity specifications for specific (raw) material need to be met for battery feedstock resale.

### 2.3.4. LIB Recycling in the U.S. and Australia

Broadly speaking, the battery recycling industry involves logistics solution providers, sorters, 4R (repair, remanufacture, refurbish, repurpose) solution providers, pre-processors, and recyclers. LIB recycling in the U.S. was initially performed by companies that focused on recycling a diverse range of products and batteries including lithium primary cells. Some of these companies are significantly growing their recycling efforts for LIBs. In addition, as the prevalence of BEVs has dramatically increased since the late 2010s, LIB recycling has become a major opportunity sector for start-ups in the U.S. Currently, there are 41 facilities working in various stages of the battery recycling value chain in the U.S., with 15 facilities planned or under construction [38]. Table 4 shows the breakdown of the facilities by value chain stage. Notable remanufacturers/repurposes, preprocessors, and recyclers are summarized in Table S3 [90–101].

**Table 4.** Number of Facilities Involved in the U.S. Battery Recycling Industry (Adapted from NAAT-Batt Database) [38].

| Activity | Current | Planned |
|---|---|---|
| Logistics | 8 | 1 |
| Sorting | 13 | 1 |
| 4Rs | 7 | 2 |
| Pre-processing | 6 | 3 |
| Recycling | 8 | 8 |
| Other | 2 | - |
| Subtotal | 41 * | 15 |

*: Numbers do not add up to 44 because 3 facilities are multi-purpose.

Along with the growth in the industry, the United States' DoE began to invest more heavily in LIB recycling, predominately through the national laboratory system, via the Critical Materials Institute and the ReCell Center. The Federal Government will also invest over AUD 7B from 2022 to 2026 via the BIL to enable sustainable sourcing and processing of battery materials as well as recycling [102]. These investments are key in creating robust supply chains for LIBs in the U.S. for the future. In the last few years, efforts by both private

industry and government have led to many companies building pilot plants and working towards full-scale recycling facilities.

Over the past decade, LIBs have been the dominant battery type in BEVs, resulting in an expected increasing supply of spent LIBs; these batteries will reach EOL approximately 12 years after introduction. Whilst there are initial discussions around reuse or second-life applications of EOL BEV batteries, there are currently insufficient quantities of these batteries and the regulations around second-life applications are still being developed and vary from country to country [68,103]. Projections based on historical BEV sales together with existing and announced battery manufacturing capacity values in the U.S. suggest that EOL BEV batteries will amount to 110,000 t in 2030; manufacturing scrap will amount to 440,000 t and will play a significant role in the viability of LIB recycling industries.

In Australia, as of 2022 [23], there are ~83,000 EVs, compared with 2.24 million in the U.S., with most yet to reach EOL. As a result, the quantity and chemistry diversity of LIB waste are currently insufficient to sustain a LIB recycling industry. Local Australian companies (Ecobatt, Victoria; Ecocycle, all states and New Zealand; and Envirostream, Victoria) currently only collect LIB waste, pre-process it to form black mass, and then export it to other countries, such as China, Korea, and Japan, according to information from the Australian Battery Recycling Institute (ABRI) [104]. However, the position on developing a domestic Australian LIB recycling industry may change in the decade to come. The dramatic uptake in photovoltaics in Australia has resulted in increased interest and activities to install residential and large-scale BESSs using LIBs. As a result, it is expected that EOL BESS LIB waste will contribute significantly more to the Australian LIB waste stream compared to the USA, where BEV waste will dominate. Other companies, such as Evolve Renewable Materials, Battery Pollution, TES-AMN, and Green Li-ion, have recently announced plans to establish LIB recycling capability in Australia. Australian companies Neometals (WA) and Pure Battery (Queensland) have developed recycling processes for LIBs but, due to the lack of suitable infrastructure and funding incentives in Australia, have partnered with German companies to develop the technology for market readiness. The Battery Stewardship Council (BSC) [105] was formed in 2022 to address the tracking, management, and coordination of the infrastructure needed across the states and territories in Australia to safely collected and transport LIB waste for recycling. Accurate information related to the number of LIBs produced, imported/exported, defective/recalled, and reaching EOL as well as the deployment of LIBs and their applications is limited in many countries due to inadequate tracking of LIBs from cradle to grave, which makes it extremely difficult to predict the number and supply of waste LIBs with any accuracy. The BSC is focused on consumer education and tracking of imported/exported and EOL LIBs in Australia to provide valuable data to stakeholders to develop sustainable LIB recycling capability within Australia [106,107].

In addition to helping address the shortages of critical chemicals used to supply the demand and growth of LIBs, changes towards an increase in the Environmental, Social, and Governance (ESG) framework, the adoption of battery passports, and a focus on greener processes (i.e., lower energy consumption, waste generation, and emissions) are contributing to changes in how LIB waste is currently being managed. The relatively low volumes of EOL LIBs in the market can hinder companies aiming to develop commercially viable recycling processes using current established industry technologies, such as pyro- and hydrometallurgy. However, reports [78,108,109] have shown that recycling LIB waste is more cost-effective and has a lower energy footprint when compared with raw materials processing, particularly for processes where battery materials can be recovered without needing to produce the pure base chemicals or form (e.g., lithium hydroxide). For example, closed-loop recycling using sustainable methods for pyro- and hydrometallurgy or the introduction of direct recycling may offer a process with lower cost, energy consumption, and waste generation, reducing the demand on virgin minerals which is highlighted in Figure 5 below [110].

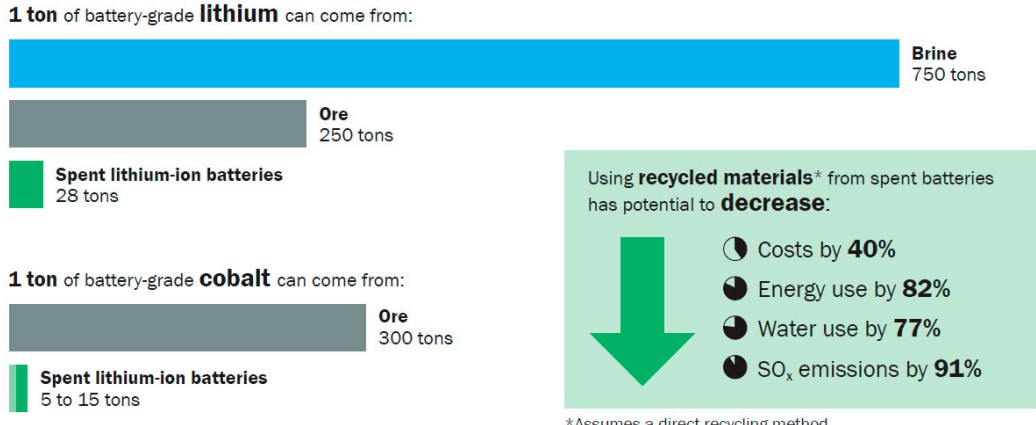

**Figure 5.** Benefits of recycling for LIBs (Reproduced from [110]).

*2.4. Other Factors and Drivers Influencing the Presence and Viability of a LIB Recycling Industry*

Towards Net Zero is focused on the balance between the amount of GHG produced and the amount that can be removed from the atmosphere [111]. Depending on the types of industries (primary, secondary, and tertiary), transportation, and consumer consumption within a country, each will have different challenges to address and, therefore, will occur at different stages of renewable energy transition [47].

2.4.1. Lowering Gas Emissions—Fuel Efficiency and Emissions Standards

The introduction of emission standards, especially for ICE vehicles, has been a tremendous driver globally towards reducing vehicle emissions. While the Euro 7 emissions standard is targeted to be implemented in 2025 [112], U.S. emission standards are slightly more ambitious than those of Europe as their standards are applicable to both petrol and diesel engines [113]. Australia, like the U.S., developed its own emission standards in the 1970s [114], but in contrast to Europe and the U.S., Australia's emission standards are only at Euro 5, which sets higher emission limits for carbon monoxide and particulate matter, and does not regulate ammonia or emissions from brakes and tires while driving. To complicate matters, Australia does not currently have a fuel efficiency standard, which regulates emission levels from new vehicles to reduce GHG and other pollutants. The current Australian Federal Government has recently announced it will consider introducing fuel efficiency standards as part of the development of Australia's first National Electric Vehicle Strategy [115]. Until then, Australia's current low fuel quality [116] means that many car manufacturers will not likely import their best low-emission (i.e., Euro 7-compliant) vehicles to Australia; [117] combined with the continued high sales of ICEs, gas emissions [118] from vehicles are likely to remain largely unchanged. The Electric Vehicle Council (EVC) of Australia recently released its report supporting the government's approach and encouraging the development a of new vehicle efficiency standard specifically for Australia [119].

2.4.2. Influence of Global and Government Policies—Lowering Emissions through Accelerated Uptake of Electric Vehicles

Recent articles [120,121] highlight that EV uptake in some countries has been significantly accelerated by government bans on new ICE vehicles or the introduction of reduction targets of the number of ICE vehicles by specific dates (Table 5). In most cases, transition to electrified mobility from ICEs has required significant government policies, regulations, and monetary incentives (e.g., in Europe, U.S., Canada, UK, Singapore, etc.) to support, educate, and initiate the change and to develop the infrastructure, industries, supply chain, and workforce needed to enable a low-emission economy. These policies also have had a "knock-on" effect where automakers have capitalized on the dramatic surge in demand for BEVs. Due to global EV shortages, exacerbated by the impact of the COVID pandemic

on global supply chains, countries like Australia without government policies towards set reduction or elimination dates for ICEs are experiencing limited supplies of BEV and PHEV brands and models and may experience long delays (i.e., years) before they are available. These delays will also mean that a reliable supply of EOL LIBs from EVs will take longer, therefore limiting the availability of waste LIBs needed to develop a sustainable LIB recycling industry. Conversely, higher EV and BESS uptake will bring greater certainty to LIB recycling industries and provide longer-term viability.

**Table 5.** Global examples of various Set Targets to Ban ICEs (Adapted from [121]).

| Country | Objectives | Target Date |
|---|---|---|
| Norway | All vehicles sold will be carbon-neutral | 2025 |
| China | 20% Vehicles are electric or hybrid<br>>50% Vehicles are electric or hybrid | 2025<br>2035 |
| Singapore | Ban on ICE cars | 2030 |
| Israel | Ban on ICE cars | 2030 |
| UK | Ban sales on ICE cars | 2030 |
| USA<br>(California) | >50% Vehicles sold are electric<br>or hybrid<br>Ban on ICE cars | 2030<br>2025 |
| Japan | Ban on sale of ICE cars | 2035 |
| India | 30% Vehicles are electric | 2035 |
| Europe<br>(Sweden, Ireland, Netherland) | Ban on sale of ICE cars and hybrid cars<br>Reach carbon neutrality | 2030<br>2035–2050 |

For instance, in February 2021, the introduction of the Singapore Green Plan 2030 [122] to support the government's aims of achieving a whole-of-nation movement and to advance Singapore's national agenda for renewable and sustainable development and economy includes a transition to electrified transport. As a result, the Green Plan has stimulated a variety of industries along the battery supply chain. In addition to a host of electrified mobility startups [123], Hyundai will commence an EV manufacturing plant in 2023 [124] and the operation of an onshore LIB recycling plant (TES-AMN group) has already commenced [125]. In contrast, a recent publication highlighting the efforts of Poland to develop onshore capability to recycle LIBs has shown that after 10 years of interest by a variety of stakeholders in the battery supply chain, it has not been able to develop, suggesting that government policies and monetary incentives play a key role [126]. Likewise, Australia's neighbours in the Asia-Pacific region are commencing their transition to renewable energy generation and storage and, therefore, opportunities exist for Australia to develop supply chains in the local region.

2.4.3. Raw and Recycled Materials: Securing Domestic and Global Supplies for Renewable Energy Growth

Countries without an abundance of natural reserves of critical and base metals, but heavily reliant on these resources to transition to renewable green energy technologies, have an urgent need to create and integrate with emerging energy supply chains to ensure domestic energy security. Some are establishing new sovereign capabilities, or partnering with those with strong trade agreements, to develop supply chains around global energy transition. The emergence of the Environmental, Social, and Governance (ESG) [127,128] framework, being developed around the battery supply chain to better ethically source and supply battery materials and manage EOL LIBs, is also creating new local industries. The development of a battery passport [129] would allow various stakeholders in the supply chain to understand the origin of materials, battery chemistry, and battery usage history. One significant benefit of the passport is that it would provide indirect value to the LIB recycling industry through a standard global identifier of LIB chemistry that is currently lacking. This would allow LIB recyclers to rapidly and easily determine EOL LIB waste streams based on chemistry and reduce some of the sorting problems the industry currently faces in returning these materials back to the supply chain.

## 3. Conclusions

*Opportunities, Risk, and Outlook*

The global transition to a net-zero economy (low-carbon-energy future) will require storage for intermittent, renewable energy sources. LIBs power EVs and are used in large-scale BESSs. The current, enormous demand for these batteries is, in turn, creating supply issues for the necessary raw materials. In general, governments can take actions to ensure a sustainable battery lifecycle, including:

- Developing regulations and policies that accelerate the transition to net-zero emissions and incentivize the growth of a transparent, ethical, and environment-friendly battery supply chain, such as the ESG framework and emission and fuel standards;
- Securing partnerships to ensure that deficits in the domestic supply chains can be reliably filled;
- Mandating methods to track batteries through their lifecycle and enable identification of EOL battery chemistry to simplify sorting and recycling;
- Creating regulation that simplifies the safe transport of EOL LIBs to recycling facilities;
- Ensuring recycling is occurring by creating policy that establishes responsibility for EOL batteries and/or making recycling sufficiently profitable;
- Mandating recycled content for new LIB production (within the physical limits);
- Developing alternative recycling technologies that are more efficient (i.e., lower energy and green chemistry) and/or batteries that use more abundant material resources.

Australia and the U.S. have taken different routes towards net-zero emissions, largely influenced by government policies and other factors relating to population density, relative abundance of primary and secondary resources, and domestic manufacturing capabilities.

The significantly lower population density in Australia and its vast, unpopulated distances between regional towns increase the logistical complexity and cost of EV infrastructure networks. Fewer government policies and financial incentives that promote EV ownership and the lack of fuel efficiency or emission standards at levels comparable to other countries have resulted in a relatively minor uptake of EVs in Australia.

More stringent ICE emission standards have been one major driver in transitioning to EVs in the U.S. Coupled with a domestic automotive manufacturing industry producing EVs and LIBs, the U.S. has a local source of battery manufacturing scrap and growing quantities of EOL batteries for recycling that will be used to produce material for new batteries.

Australia has been active in deploying renewable energy generation with BESSs to reduce emissions and achieve energy security. Although the cumulative BESS capacity in the U.S. is small compared to EVs, it has quadrupled over the past two years. As BESSs are being developed in both countries at a rapid rate to utilize the large number of solar-, wind-,

and hydroelectric-based technologies being installed, this is potentially an area in which the two countries could collaborate to further increase the deployment of these energy generation and storage technologies.

With government policies related to sovereign (manufacturing) capability and decarbonization, the U.S. is expected to have sufficient manufacturing scrap and EOL batteries from EVs to enable a domestic recycling industry and potentially process material from other countries. Australia, on the other hand, may not be able to support an industry from its own EOL material, even from its large-scale BESS operations. Options for Australia could include becoming a regional recycling service provider (i.e., by accessing EOL batteries from neighbouring countries) or developing new technologies for processing EOL batteries alongside beneficiated ores as part of their growing refining industry. The U.S. already has EVs and LIB manufacturing capabilities, and it is slowly building capacity in the production of battery active materials and precursors, raw material extraction and refining, and recycling. Although sufficient natural resources of critical minerals for battery manufacture are not currently available onshore, the U.S. could explore versatile technologies that could process both recycled and virgin feedstocks and/or unconventional resources such as mine tailings and sea nodules. Australia, on the other hand, lacks an EV industry but has nascent battery manufacturing capability. However, it is rich in critical mineral resources and is building domestic processing facilities to increase the value of its resources by refining the ores to produce battery materials, which it may either use locally or export. There is significant potential synergy for a collaboration in battery material production that could accelerate activities in both countries. A collaboration based on a mutually beneficial relationship could enable exchange in processing and manufacturing technology and battery materials along the value chain. It is also important for both countries to develop their own battery supply chains, to further secure their critical material supplies and reduce transportation costs and emissions. The Australian supply chain could be built from virgin resources and the U.S. supply chain could be built from recycled feedstock, and the supply chain of both countries could be built from potentially unconventional resources, offering valuable insights for other regions of the world who also aspire to develop a resilient and sustainable supply chain for clean energy transition.

The impact, influence, and progress made by U.S. government policies and incentive, business, industry, and community experiences could help inform Australia as they develop their own strategies, establish new sustainable industries in the battery supply chain, and increase EV uptake and infrastructure. As the U.S. rapidly ramps up its large-scale BESS capacity and integration with renewable energy generation, it could collaborate with Australia for battery materials for U.S. battery production plants. As both countries continue to progress along their individual pathways towards Net Zero, there are numerous opportunities for shared learnings which will deliver a sustainable energy future for the two nations for global benefit.

Much of the information examined in this paper is applicable to other countries seeking to develop onshore LIB recycling and battery value chains. Global battery value chains are changing and evolving rapidly because of increasing demand, growth of new industries, changes to government regulations, and the increasing supply of waste LIBs; these aspects are worth investigating by other countries. It would be worthwhile to revisit progress on the LIB battery chain and recycling in both Australia and U.S. in 5–10 years from now to understand what has worked and what has not. At the same time, other countries could benefit from looking at their current processes for managing EOL LIBs and whether recycling can occur onshore or requires processing in neighbouring countries. Where and when these global LIB supply chains and recycling industries are brought online will heavily impact other countries' efforts to develop similar industries onshore.

**Supplementary Materials:** The following supporting information can be downloaded at: https://www.mdpi.com/article/10.3390/recycling8050078/s1, Figures S1–S7 and Tables S1–S3. It contains the materials and methodology used in the preparation of manuscript. Reference [130] is cited in the supplementary materials.

**Author Contributions:** G.E.C. was involved the conceptualisation, methodology, data curation, investigation, visualization, funding acquisition, writing, reviewing, and editing of the original draft preparation. Q.D. was involved in conceptualisation, data curation, investigation, writing, reviewing, and editing of the original draft preparation. J.S.C.L. was involved in data curation, writing, reviewing, and editing of the original draft preparation. A.L. was involved in conceptualisation, data curation, investigation, writing, reviewing, and editing of the original draft preparation. L.G. was involved in conceptualisation, writing, reviewing, and editing of the original draft preparation. Y.Z. was involved data curation, investigation, visualization, writing, reviewing, and editing of the original draft preparation. J.S. was involved in conceptualisation, funding acquisition, writing, reviewing, and editing of the original draft preparation. All authors have read and agreed to the published version of the manuscript.

**Funding:** This research was co-funded by the CSIRO Manufacturing Business Unit (Australia) and CSIRO Global USA. This work was co-authored by Argonne National Laboratory, a U.S. Department of Energy laboratory managed by UChicago Argonne LLC under contract DE-AC02-06CH11357.

**Data Availability Statement:** Not applicable.

**Acknowledgments:** CSIRO would also like to thank the CEO (Katharine Hole) of the Australian Battery Recycling Institute (ABRI), Toby Hagon (Editor, EV Central, Australia), the Electric Vehicle Council, Australia, and Naomi Boxall (CSIRO Environment Business Unit) for fruitful discussions. We thank Naomi Boxall for critical review and feedback on the manuscript. All authors' efforts were supported by the Vehicle Technologies Office, Office of Energy Efficiency and Renewable Energy, United States Department of Energy. We would like to thank, Tina Chen, Jake Herb, and Brian Cunningham from the Vehicle Technologies Office, Office of Energy Efficiency and Renewable Energy, United States Department of Energy, for their guidance and support.

**Conflicts of Interest:** The authors declare no conflict of interest.

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
