# Peer review of "Closing the Loop on LIB Waste: A Comparison of the Current Challenges and Opportunities for the U.S. and Australia towards a Sustainable Energy Future"

_recycling, doi:10.3390/recycling8050078_

Round 1
Reviewer 1 Report
This paper focus on a topic of great interest. Interesting data has been provided and a comparison is made that may be valuable for the future. The graphs and figures that have been used in the article should also be highlighted.
At the same time, some points are observed that the authors should review. Firstly, the description of the methodology that has been used to carry out this research is missing. The authors have not explained what methodology they have followed to achieve the objective of the research. The structure that has been followed in the article is not briefly exposed either, in fact, nor is it mentioned. In addition, the introduction presents some data from the two countries compared: the United States and Australia, however, it is not sufficiently justified why it is worth comparing these two realities. Likewise, the data from the census of the United States and Australia is given, however, in the case of the United States, the year 2020 is taken as the source, while in the case of Australia, the year 2021 is used. It would be convenient to use the same year as it is a comparison. According to the authors, this article discusses perspectives that influence “the barriers, technical challenges, opportunities, and risks for LIB recycling. This understanding can help diverse stakeholders plan and implement solutions to address LIB recycling and battery supply 40 chain gaps in both nations”.However, it is overlooked that in point 5, in which other factors and drivers are addressed, these stakeholders are not mentioned and remain focused on industry and government. It would have been interesting to talk about the interest/sensitivity of consumers or citizens, or the role of the media in this area, or to specify which stakeholders this research hopes to contribute to.
Author Response
Dear Reviewer 1 and Editor,
I thank the reviewer for the time reading and providing feedback on our manuscript to improve the quality of the manuscript.
Below are the reviewers points and how we have addressed them using track changes.
- Editor – adjust this paper structure to “Introduction, Results, Discussion, Materials and Methods and Conclusion”.
- These key headings have been added to the manuscript, along with subheadings.
- Description of the methodology that has been used to carry out this research is missing. The authors have not explained what methodology they have followed to achieve the objective of the research. In addition, the introduction presents some data from the two countries compared: the United States and Australia, however, it is not sufficiently justified why it is worth comparing these two realities.
- The authors have included a section in the supplementary information section titled Methodology. As it is unlike a typical technical paper, it provides background as to why the two countries and these national laboratories have engaged in this discussion and chosen to publish the work as it is relevant to many other countries looking to develop a battery value chain and LIB recycling industry.
- Likewise, the data from the census of the United States and Australia is given, however, in the case of the United States, the year 2020 is taken as the source, while in the case of Australia, the year 2021 is used. It would be convenient to use the same year as it is a comparison.
- As mentioned above, unlike a typical technical paper, the frequency of reports, media press releases, government policies/regulation changes/updates, publications are occurring at a rapid rate and therefore it is difficult to include, and always access, the most recent articles. The authors have also attempted to choose relevant articles, given vast amount of literature available.
- Were possible, we have attempted to compare information in a similar timeframe, but through discussions with various stakeholders this information is changing and sometimes not readily accessible to the public due to commercial constraints. Thus, for the purpose of initiating a discussion on the topic and to have those in the community consider and provide further discussion, it is important to get what information we can access presented to the community.
- To assist in the introduction and throughout the paper, we have added more references that have recently been published and others that are also relevant.
- According to the authors, this article discusses perspectives that influence “the barriers, technical challenges, opportunities, and risks for LIB recycling. This understanding can help diverse stakeholders plan and implement solutions to address LIB recycling and battery supply 40 chain gaps in both nations”. However, it is overlooked that in point 5, in which other factors and drivers are addressed, these stakeholders are not mentioned and remain focused on industry and government. It would have been interesting to talk about the interest/sensitivity of consumers or citizens, or the role of the media in this area, or to specify which stakeholders this research hopes to contribute to.
- As an example, in Australia CSIRO has and continues to work closely with the Australian Battery Stewardship Initiative (ABRI) and the Battery Stewardship Council (BSC) Program. Here these organisation are gathering information from other stakeholders that we have taken into consideration when reporting information in the text and have added references of their reports accordingly. These organisations have outward programs to engage with media and citizens to raise the awareness of LIB use and waste management at end-of-life at a state government level. In contrast, in the U.S. much of the activities of the LIB value chain and recycling is drive by the Federal Government through the Department of Energy and delivered through the many national laboratories in the US.
- As both ANL and CSIRO are R&D organisations, the focus has been on how these “the barriers, technical challenges, opportunities, and risks for LIB recycling” can be improved through discussions and understanding the challenges with industry, government and academic institutions.
The authors have used track changes to show where these changes have been made in the manuscript and SI and therefore have addressed the points raised by the reviewer.
Reviewer 2 Report
I have read the manuscript “Closing the Loop on LIB Waste: A Comparison of the Current Challenges and Opportunities for U.S. and Australia Towards a Sustainable Energy Future” submitted to Recycling, MDPI.
The subject of this manuscript is interesting and the authors display valuable information on the challenges, fences, prospects, and experience of the United States of America and Australia into management of renewable energy storage and development of battery supply chain for lithium-ion batteries. The document is comprehensive, the discussion is reasonable, meanwhile, there are some corrections/clarifications need to be made for the current document. So, I think this document should be considered for publication after minor modifications. Some of my specific comments are below:
Point 1. Abstract is general, I recommend the authors edit it to make it more concrete and specific, for example some measurable data could be included.
Point 2. For the sentence “As such, critical metals and chemical”, The authors should provide some examples of some of these metals/chemical” for a better understanding of the document.
Point 3. Abstract. Lines 16-18, for the sentence “Dramatic increases in raw material prices, coupled with. LIBs degrade over time”. Please, avoid generalizations and provided quantitative evidence, I think that this sentence could be improved.
Point 5. Abstract. I considered that the authors use I think that authors use many keywords, they could consider using the most relevant ones.
Point 6. Introduction. The reason for only using information from the United States and Australia should be made clear.
Point 7. Conclusions section. Please, comment on how the experiences of the US and Australia can be applied to other regions of the world.
Author Response
Dear Reviewer 2 and Editor,
I thank the reviewer for their time reading and providing feedback on our manuscript to improve the quality of the work. The authors thank the reviewer for their positive feedback in all the criteria that the manuscript has been deemed suitable for publication with minor changes.
Below are reviewers comments and how we have addressed these points.
- Point 1. Abstract is general, I recommend the authors edit it to make it more concrete and specific, for example some measurable data could be included.
- The authors have included some data from the manuscript into the abstract as suggested by Reviewer.
- Point 2. For the sentence “As such, critical metals and chemical”, The authors should provide some examples of some of these metals/chemical” for a better understanding of the document.
- This has been addressed in the point above.
- Point 3. Abstract. Lines 16-18, for the sentence “Dramatic increases in raw material prices, coupled with. LIBs degrade over time”. Please, avoid generalizations and provided quantitative evidence, I think that this sentence could be improved.
- As in the Point 1 additional information has been added to provide quantifiable examples.
- Point 5. Abstract. I considered that the authors use I think that authors use many keywords, they could consider using the most relevant ones.
- The authors have reduced the number of keywords from 12 to 9 words.
- Point 6. Introduction. The reason for only using information from the United States and Australia should be made clear.
- The authors have added a section in the supplementary section under Methodology explaining why the two national laboratories, CSIRO and ANL, have chosen to work together on this topic.
- Point 7. Conclusions section. Please, comment on how the experiences of the US and Australia can be applied to other regions of the world.
- The authors have included some information around this in the Methodology section in the Supplementary Section and also another paragraph in the conclusion section.
The authors have used track changes to show where these changes have been made in the manuscript and SI and therefore have addressed the points raised by the reviewer.